

# Clinical diagnosis, treatment, and survival analysis of 61 cases of salivary duct carcinoma: a retrospective study

Shubin Dong[1], Mengru Li[2], Zhiwei Zhang[3], Bowei Feng[4], Wei Ding[1], Jiang Chang[5] and Feng Liu[1]

[1] Department of Head and Neck Surgery, Shanxi Provincial Cancer Hospital, Shanxi Hospital Affiliated to Cancer Hospital, Chinese Academy of Medical Sciences, Cancer Hospital Affiliated to Shanxi Medical University, Taiyuan, Shanxi Province, China
[2] Academy of Medical Sciences, Shanxi Medical University, Taiyuan, Shanxi Province, China
[3] First Clinical Medical School, Shanxi Medical University, Taiyuan, Shanxi Province, China
[4] School of Stomatology, Shanxi Medical University, Taiyuan, Shanxi Province, China
[5] Shanxi Provincial Cancer Hospital, Department of Pathology, Taiyuan, Shanxi, China

Corresponding author
Feng Liu, liufeng@sxmu.edu.cn

## ABSTRACT

**Objective.** This study aims to investigate the clinicopathological features, treatment modalities, and prognostic factors associated with salivary duct carcinoma (SDC).

**Methods.** A retrospective analysis was conducted on the clinicopathological data of 61 patients with SDC admitted to Shanxi Cancer Hospital from January 2010 to February 2024. This study focused on their demographic information, treatment regimens, clinical outcomes, and overall prognosis.

**Results.** A total of 61 patients with SDC were included in this study, of whom 45 (73.77%) had primary tumors in the parotid gland, 44 (72.13%) were in stage IV at the initial visit, 35 (57.38%) had cervical lymph node metastases, and two (3.28%) had distant metastasis at the first visit. Immunohistochemical (IHC) staining showed that 54 cases (88.52%) were positive for androgen receptor (AR), and 25 cases (40.98%) were positive for human epidermal growth factor receptor 2 (HER-2). Two patients did not undergo surgical treatment due to distant metastasis and received palliative chemoradiotherapy. Fifty-nine patients underwent radical surgery, and of these, 52 (88.14%) received postoperative adjuvant treatment, including chemoradiotherapy or radiotherapy alone. The median overall survival (OS) time was 20 months for the non-surgery group ($n = 2$), 79 months (95% CI [72.60–85.40]) for the surgery and chemoradiotherapy group ($n = 22$), 61 months (95% CI [49.11–72.90]) for the surgery and radiotherapy group ($n = 30$), 20 months (95% CI [13.60–26.40]) for the surgery-only group ($n = 7$). The median OS was significantly higher in the adjuvant treatment groups (radiotherapy or chemoradiotherapy) compared with the surgery only or non-surgery groups ($P < 0.05$).

**Conclusion.** SDC is a rare and highly aggressive malignancy with a poor prognosis. Surgical margins, American Joint Committee on Cancer (AJCC) stage, and treatment method were identified as adverse prognostic factors affecting the OS of patients with SDC. Radical surgery remains the primary treatment for salivary duct carcinoma while adjuvant therapy significantly reduces recurrence rates and improves survival. However, the overall prognosis remains challenging, highlighting the need for novel therapeutic strategies.

## INTRODUCTION

Salivary duct carcinoma (SDC) is a highly aggressive tumor that arises from the epithelial cells of the salivary duct excretory system and may develop *de novo* or from a preexisting pleomorphic adenoma (SDC ex-PA). In 1968, *Kleinsasser, Klein & Hübner (1968)* first described SDC, noting its morphological similarity to high-grade ductal carcinoma of the breast, with both invasive and intraductal components, leading to its designation as "salivary duct carcinoma". SDC was officially recognized as a distinct tumor type in the World Health Organization (WHO) classification of salivary gland tumors (*WHO, 2022*). SDC accounts for approximately 5%–10% of all salivary gland carcinomas (SGCs) (*Lépine, 2024*; *Tchekmedyian, 2021*). This malignancy predominantly affects middle-aged and older men, and most commonly occurs in the parotid gland, followed by the submandibular gland, sublingual gland, and other areas. By the time most patients seek medical attention, the tumor is typically at an advanced stage (*Nachtsheim et al., 2023*; *Nakaguro et al., 2020*), with 47%–73% of patients presenting with cervical lymph node metastasis (*Schmitt, Kang & Sharma, 2017*; *Zhang & Li, 2023*). Currently, the standard treatment for SDC mirrors that of other high-grade SGCs, involving radical resection, lymph node dissection, and postoperative adjuvant radiotherapy and chemotherapy (*Van Herpen et al., 2022*). No specific systemic treatment is available for the recurrence or metastasis of SDC. However, for cases with androgen receptor (AR) or human epidermal growth factor receptor 2 (HER-2) expression or alterations in other molecular pathways, personalized therapies may be considered (*Mayer et al., 2024*). Despite surgery and postoperative adjuvant therapy, the overall prognosis for patients with SDC remains poor. Local recurrence or regional lymph node metastasis occurs in 27%–74% of patients, and 9.8%–40.0% experience distant metastasis (*Stodulski et al., 2019*; *Zhang & Li, 2023*). The survival rate for SDC is relatively low, with approximately half of patients succumbing within 5 years of diagnosis (*Hirai et al., 2023*; *Laughlin et al., 2023*).

While surgery and adjuvant therapy form the cornerstone of management, the integration of molecular targeted therapies represents a promising frontier. This study retrospectively analyzed the clinicopathological data of 61 patients diagnosed with SDC. By incorporating immunohistochemistry (IHC) and fluorescence *in situ* hybridization (FISH), we identified AR and HER-2 as potential therapeutic targets in SDC. This study contributes to the field by providing a comprehensive analysis of the clinicopathological features and prognostic factors in a relatively large cohort of SDC patients, and by highlighting the potential benefits of molecularly guided therapies in improving patient outcomes. Furthermore, our study aims to address the urgent need for novel treatments and improved patient survival by investigating the clinical outcomes of personalized therapies based on molecular markers such as AR and HER-2.

## MATERIALS AND METHODS

### Clinicopathological data

In this retrospective cohort study, data were collected from patients diagnosed with SDC at the institution between January 2010 and February 2024. In 2010, our hospital transitioned to a new electronic medical record (EMR) system. Accessing patient records from before this change proved to be inconvenient. Therefore, prior to commencing the study, we opted to focus on the period spanning from 2010 to 2024. In this study, we conducted pathological and IHC examinations on all samples according to the 2022 WHO classification criteria to meet the diagnostic standards for SDC. Histological evaluation focused on several key features:

Cellular morphology: assessment of cellular pleomorphism, nuclear atypia, and mitotic activity. Architectural patterns: identification of tubular, cribriform, and solid growth patterns typical of SDC. Necrosis: presence and extent of tumor necrosis. Invasion: evidence of perineural invasion (PNI) and lymphovascular invasion (LVI). Additionally, immunohistochemical staining was performed to detect the expression of AR, HER-2, GATA3, and GCDFP-15. During the pathological diagnosis process, samples with cytological features suggestive of SDC on hematoxylin-eosin (HE) stained sections routinely underwent IHC testing for AR and HER-2. For patients with a HER-2 score of 2+, FISH testing was performed to confirm gene amplification.

IHC scoring: HER-2 protein expression was scored based on staining intensity and the proportion of stained cells.

The scoring criteria were as follows:

0: No staining or weak membranous staining, with < 10% of cells stained.

1+: Weak to moderate membranous staining, with $\geq$ 10% of cells stained.

2+: Strong membranous staining, with $\geq$ 10% of cells stained but heterogeneous staining.

3+: Strong membranous staining, with $\geq$ 10% of cells stained and homogeneous staining.

FISH Testing for HER-2 Gene Amplification:

FISH testing was used to detect HER-2 gene amplification, with a HER-2/CEP17 ratio $\geq$ 2.0 indicating gene amplification.

AR status assessment: AR status was assessed using IHC to detect AR protein expression levels, evaluated using the H-score system (range 0–300).

0: No staining.

1–100: Weak staining.

101–200: Moderate staining.

201–300: Strong staining.

Positive standard: Patients with an H-score $\geq$ 100 were considered AR positive.

This study was reviewed and approved by the Medical Ethics Committee of Shanxi Cancer Hospital (ethics approval number: KY2024081). The study was conducted in accordance with the Declaration of Helsinki, and all participants, or their legal guardians, provided written informed consent after fully understanding the purpose of the study.

## Inclusion and exclusion criteria

The inclusion criteria for this study were as follows: patients with SDC originating from the major salivary glands (parotid, submandibular, and sublingual glands) and minor salivary glands in the oral cavity (labial, palatal, and buccal glands) were included. Both *de novo* and SDC ex-PA cases were considered. Patients were required to have pathological and IHC diagnoses consistent with SDC. Additionally, patients with complete medical records and a follow-up period exceeding six months were included in the study.

The exclusion criteria were as follows: patients whose first SDC resection was performed at another hospital were excluded. Patients with a history of or current diagnosis with other types of malignant tumors were also excluded. Furthermore, patients or their families who refused surgical treatment as the first-line therapy after pathological confirmation of the biopsy results were excluded from the study. Lastly, patients with missing clinical and pathological data or incomplete follow-up information were not included in the analysis.

## Preoperative auxiliary examination and treatment plan

All patients underwent a preoperative assessment that included a complete blood count, liver and renal function tests, and neck ultrasound to evaluate the tumor staging. Additionally, in our study, the decision to perform further imaging studies such as neck computed tomography (CT) or magnetic resonance imaging (MRI) was based on several clinical factors, including suspicious findings on initial ultrasound (*e.g.*, unclear or suspicious results regarding lymph node metastasis or tumor invasion), clinical symptoms suggestive of advanced disease (*e.g.*, pain, rapid growth of the mass, facial nerve paralysis, or other signs indicative of advanced or invasive tumor behavior), and suspected distant metastasis (based on patient symptoms or laboratory findings). Ultrasound, CT, and/or MRI were combined to comprehensively assess tumor characteristics and guide treatment decision (*D'Heygere, Meulemans & Vander Poorten, 2018*; *Kim et al., 2019*; *Weon et al., 2012*). The treatment plan was collaboratively discussed by a multidisciplinary team and carried out with the informed consent of the patient and their family. All cases were operated on by surgeons experienced in salivary gland tumor surgery. Preoperative imaging examinations were used to evaluate the tumor's size, involvement of surrounding tissues and adjacent organs, nerve invasion, and the presence of distant metastasis. If no distant metastasis was detected and the tumor was considered resectable, radical surgical resection was performed. The extent of resection was determined by the surgeon, based on a comprehensive assessment of the tumor's specific characteristics during surgery, including local tumor infiltration, lymph node metastasis, nerve invasion, and other relevant factors, as well as the patient's overall condition. During the operation, frozen section pathology was routinely performed to determine the extent of surgical resection and to ensure negative surgical margins. If distant metastasis was identified during the preoperative assessment, palliative antitumor treatment was administered according to the patient's condition. The implementation of the relevant treatment plans was carried out with the informed consent of the patients and their families, and an informed consent form was signed.

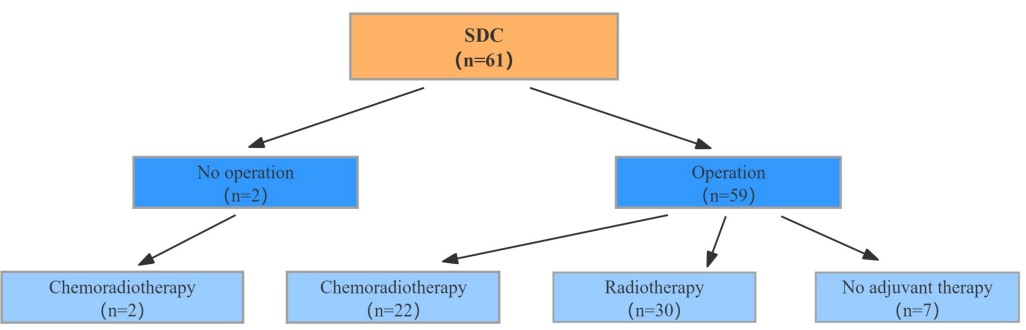

**Figure 1** Treatment strategies in patients diagnosed with SDC.

## Treatment strategies

The specific treatment strategies for the 61 patients with SDC were illustrated in Fig. 1. Of the patients included in this study, two did not undergo radical surgery because of distant metastasis and instead received palliative chemoradiotherapy. In contrast, 59 patients underwent radical surgery. The decision to administer adjuvant therapy was based on a comprehensive assessment of several clinical and pathological factors. These included surgical margin status, tumor stage, evidence of lymph node metastasis, and nerve invasion. Patients with positive surgical margins, advanced tumor stages (American Joint Committee on Cancer (AJCC) III-IV), or lymph node metastasis were more likely to receive adjuvant radiotherapy or chemoradiotherapy to reduce the risk of recurrence and improve survival outcomes. Additionally, the treatment plan was tailored to individual patient characteristics such as age, physical condition, and pathological report findings. Among the 59 patients who underwent surgery, 30 received adjuvant radiotherapy, 22 received adjuvant chemoradiotherapy, and seven did not receive any adjuvant treatment.

## Follow-up

During the first 2 years after surgery, follow-up were conducted every 3 months, and then every 6 months from the third year onward. Re-examinations encompassed both imaging and laboratory examinations. Throughout the follow-up period, disease recurrence was confirmed through imaging or histopathological examination. A dedicated research nurse conducted monthly telephone follow-ups to ascertain the patient survival status. For patients who did not return to the outpatient clinic for regular follow-up visits, the dedicated research nurse conducted telephone interviews and recorded the relevant follow-up results. The dataset was initially compiled in February 2024. To ensure the completeness and accuracy of follow-up data, a final follow-up was conducted in August 2024, during which we updated the survival status and other relevant outcomes for all patients included in the study.

## Statistical analysis

Statistical analysis was performed using SPSS 27.0 (IBM Corp., Armonk, NY, USA) and R 4.4.1 software. Continuous variables were expressed as mean ± standard deviation, and categorical variables were presented as numbers (percentages). Overall survival (OS) was

defined as the period from the date of surgery to death from any cause or the date of the last follow-up. Disease-free survival (DFS) was defined as the period from the date of radical surgery to tumor recurrence, metastasis, or the date of the last follow-up. The normal distribution of continuous variables was verified using the Shapiro–Wilk test. Kaplan–Meier curves were generated to visualize survival rates, and statistical analyses were conducted to calculate survival probabilities. The log-rank test was used to compare survival curves between different groups. The covariates underwent an initial screening using Cox regression. Only the variables with $p$-value $< 0.05$ in the univariate analysis were included in the subsequent multi-factor Cox regression. Statistical significance was set at $P < 0.05$.

## RESULTS

### Basic information and treatment strategies for patients with SDC

The clinical and pathological data of the 61 patients with SDC are summarized in Table 1.

Clinical and epidemiologic characteristics: The study included 61 patients with SDC, with a mean age of $59.62 \pm 11.22$ years and a male-to-female ratio of approximately 5:1. The most common primary tumor location was the parotid gland (73.77%). Common clinical symptoms included a rapidly enlarging mass in the parotid gland or submandibular area, with some patients presenting with symptoms of nerve invasion such as facial nerve palsy or abnormal oral/facial sensation. Preoperative clinical TNM staging indicated that 52 patients (85.24%) were in stage III or IV.

Pathologic characteristics: The growth patterns SDC including glandular, linear, papillary, cribriform, and solid arrangements. The majority of SDC cases exhibited glandular and solid patterns. Necrosis within the tumor tissue was also observed under the microscope. In this study, 32 out of 61 SDC patients (52.46%) had necrosis, with 20 cases (32.79%) showing comedo necrosis. Histological evaluation revealed that 49 patients (80.33%) had *de novo* SDC and 12 (19.67%) had SDC exPA. Intraoperative exploration and postoperative pathology revealed PNI in 32 patients (54.24%), with facial nerve involvement in 21 cases, lingual nerve involvement in five cases, combined lingual and hypoglossal nerve involvement in two cases, and inferior alveolar nerve invasion in four cases. LVI was observed in 20 (33.90%) patients. Graded based on histological features, 52 cases (85.25%) of SDC were classified as G3 (high grade), while nine cases (14.75%) were classified as G2 (intermediate grade). Positive surgical margins were observed in 17 patients (28.81%), and extracapsular extension was observed in 38 patients (64.41%). Lymph node metastasis occurred in 35 patients (57.38%), and distant metastasis was present in three patients (4.92%). IHC staining showed positivity for AR in 54 cases (88.52%) and HER-2 in 25 cases (40.98%). HER-2 gene amplification was confirmed in eight cases using FISH. In this study, the median Ki-67 index was 60% (range: 10%–85%), further indicating the high invasiveness of SDC. For malignant tumors, pathologists routinely assess capsular invasion, seven out of the 12 SDC ex-PA cases (58.33%) exhibited capsular invasion.

Treatment, tumor behavior, and follow-up: Two patients with distant metastasis received palliative chemoradiotherapy, while 59 underwent radical surgery. Of those who

**Table 1  Clinical and pathological characteristics of 61 SDC patients.**

|  | $n = 61$ |
|---|---|
| Age (years, X ± S) | 59.62 ± 11.22 |
| Sex (male:female) | 50:11 |
| Tumor location (*n*, %) |  |
|     Parotid gland | 45 (73.77) |
|     Submandibular gland | 14 (22.95) |
|     Sublingual gland | 2 (3.28) |
| T (*n*, %) |  |
|     T1 | 2 (3.28) |
|     T2 | 11 (18.03) |
|     T3 | 13 (21.31) |
|     T4 | 35 (57.38) |
| N (*n*, %) |  |
|     Nx | 21 (34.43) |
|     N0 | 7 (11.48) |
|     N1 | 5 (8.20) |
|     N2 | 20 (32.78) |
|     N3 | 8 (13.11) |
| M (*n*, %) |  |
|     M0 | 58 (95.08) |
|     M1 | 3 (4.92) |
| AJCC staging (*n*, %) |  |
|     Stage I | 2 (3.28) |
|     Stage II | 7 (11.48) |
|     Stage III | 8 (13.11) |
|     Stage IV | 44 (72.13) |
| Pathological type (*n*, %) |  |
|     SDC *de novo* | 49 (80.33) |
|     SDCexPA | 12 (19.67) |
| PNI (*n*, %) | 32 (54.24) |
| LVI (*n*, %) | 20 (33.90) |
| Positive margin (*n*, %) | 17 (28.81) |
| ECE (*n*, %) | 38 (64.41) |
| Treatment strategy (*n*, %) |  |
|     Surgery only | 7 (11.48) |
|     Surgery-chemoradiotherapy | 22 (36.06) |
|     Surgery-radiotherapy | 30 (49.18) |
|     Palliative radiochemotherapy | 2 (3.28) |
| Status at last follow-up (*n*, %) |  |
| Neoplastic death | 25 (40.98) |
| Non-neoplastic death | 3 (4.92) |
| Disease-free survival | 25 (40.98) |
| With disease survival | 8 (13.12) |

**Notes.**
Abbreviation: SDCexPA, SDC ex pleomorphic adenoma; PNI, perineural invasion; LVI, lymphovascular invasion; ECE, extra-capsular extension.

**Table 2  Surgical Information of 59 SDC patients.**

| Tumor location | Parotid superficial lobectomy | Parotid superficial lobectomy + lymph node dissection | Total parotidectomy | Total parotidectomy + lymph node dissection | Radical resection | Radical resection + lymph node dissection | Total |
|---|---|---|---|---|---|---|---|
| Parotid gland | 10 | 7 | 9 | 19 | 0 | 0 | 45 |
| Submandibular gland | 0 | 0 | 0 | 0 | 0 | 13 | 13 |
| Sublingual gland | 0 | 0 | 0 | 0 | 0 | 1 | 1 |
| Total | 10 | 7 | 9 | 19 | 0 | 14 | 59 |

had surgery, 52 (88.14%) received postoperative adjuvant treatment (chemoradiotherapy or radiotherapy alone). During the median follow-up time of 64 months, 25 patients experienced neoplastic death, three died from non-neoplastic causes, 25 were disease-free, and eight were surviving with disease.

The detailed surgical information of the 59 patients who underwent radical resection is summarized in Table 2; none of these patients died during hospitalization or within 30 days post-surgery.

## Survival analysis

### The clinical, pathological, and survival analysis of all cases

Statistical analysis of the 61 patients with SDC included in this study showed that median OS did not statistically differ across patients with varying sex, age, pathological type, PNI, ECE, or LVI (all $P > 0.05$). However, statistically significant differences in median OS were observed among patients with different surgical margins ($P < 0.01$), AJCC stages ($P = 0.035$), and treatment methods ($P < 0.001$), as detailed in Table 3. Log-rank tests identified factors influencing prognosis, and a Cox regression model was constructed. Results indicated that surgical margin status, AJCC staging, and treatment modality are independent prognostic factors for SDC ($P = 0.049$, $0.047$, and $0.001$, respectively), as summarized in Table 4.

Among the 61 patients, the median follow-up time was 64 months and the median OS was 63 months (95% CI [40.13–85.87]), the survival curves are shown in Fig. 2A. These patients were divided into four groups based on the treatment modality. The median overall survival time was 20 months for the non-surgical group ($n = 2$), 79 months (95% CI [72.60–85.40]) for the surgery-chemoradiotherapy group ($n = 22$), 61 months (95% CI [49.11–72.90]) for the surgery-radiotherapy group ($n = 30$), and 20 months 95% CI [13.60–26.40]) for the surgery-only group ($n = 7$). The survival curves of patients treated with different regimens are shown in Fig. 2B. The difference in median OS between the surgery-radiotherapy/surgery-chemoradiotherapy groups and surgery-only/non-operation groups was statistically significant ($P < 0.001$).

### The clinical, pathological, and survival analysis of parotid cases

The parotid gland was the primary focus of this study due to its higher incidence of SDC compared to other salivary glands. Approximately 73.77% of the cases in this study

**Table 3   Comparison of median surviv al time among 61 SDC patients with different clinical and pathological features.**

| Clinical and pathological features | | Median overall survival time (month) | 95% CI | | P value | Log-rank |
|---|---|---|---|---|---|---|
| | | | Lower | Upper | | χ2 |
| Sex | Male | 74 | 53.97 | 94.03 | 0.973 | 0.68 |
| | Female | 51 | – | – | | |
| Age | <60 | 74 | 50.69 | 97.31 | 0.644 | 0.20 |
| | ≥60 | 61 | 35.99 | 86.01 | | |
| Surgical margin | Negative | 79 | 55.94 | 102.06 | <0.01 | 6.83 |
| | Positive | 28 | 12.55 | 43.45 | | |
| AJCC staging | I+II | – | – | – | 0.035 | 4.32 |
| | III+IV | 61 | 46.42 | 75.58 | | |
| Pathological type | SDC exPA | 56 | 31.25 | 80.75 | 0.082 | 3.14 |
| | SDC *de novo* | 33 | 14.85 | 51.15 | | |
| PNI | Yes | 61 | 36.16 | 85.84 | 0.212 | 1.53 |
| | No | 79 | 28.42 | 85.84 | | |
| LVI | Yes | 37 | 15.92 | 58.08 | 0.330 | 1.01 |
| | No | 74 | 54.87 | 93.12 | | |
| ECE | Yes | 41 | 17.26 | 64.74 | 0.247 | 1.35 |
| | No | 64 | 39.35 | 88.65 | | |
| Treatment strategy | Surgery-chemoradiotherapy | 79 | 72.60 | 85.40 | | 12.34 |
| | Surgery-radiotherapy | 61 | 49.11 | 72.89 | | |
| | Palliative radiochemotherapy | 20 | – | – | <0.001 | |
| | Surgery only | 20 | 13.60 | 26.40 | | |

**Notes.**

Abbreviation: SDCexPA, SDC ex pleomorphic adenoma; PNI, perineural invasion; LVI, lymphovascular invasion; ECE, extracapsular extension.

**Table 4   Multivariate Cox regression analysis of 61 SDC patients.**

| Factors | b-value | SE (β) | Wald | P value | Exp (B) | 95% CI |
|---|---|---|---|---|---|---|
| Surgical margin | −0.826 | 0.49 | 2.842 | 0.049 | 0.438 | 0.17–1.14 |
| AJCC staging | 1.58 | 0.79 | 4.000 | 0.047 | 4.75 | 1.02–22.12 |
| Treatment strategy | 1.32 | 0.53 | 19.36 | 0.001 | 3.74 | 1.31–10.69 |

originated in the parotid gland, making it the most common site for SDC. This focus allows for a more detailed analysis and understanding of SDC in the most prevalent location. Statistical analysis of the 45 patients with parotid SDC included in this study showed that median OS did not statistically differ based on sex, age, PNI, LVI, ECE, or pathological type (all $P > 0.05$). However, statistically significant differences in median OS were observed related to surgical margin ($P = 0.003$), AJCC stage ($P < 0.001$), treatment method ($P = 0.002$), and lymph node dissection status ($P = 0.049$), as detailed in Table 5. Log-rank tests identified factors influencing prognosis in 45 SDC patients, and a Cox regression model was constructed. Results indicated that all were independent predictors of

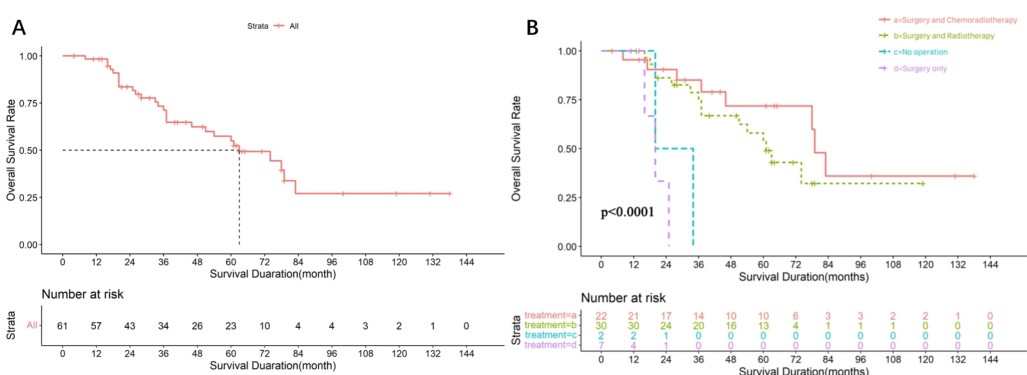

**Figure 2** **Survival analysis of 61 SDC patients.** (A) Overall survival rate of 61 SDC patients. (B) Overall survival rate of 61 SDC patients with different treatment regimens.

OS in patients with parotid SDC ($P = 0.015, 0.011, 0.009, 0.04$, respectively), as summarized in Table 6.

All 45 patients with parotid SDC cases underwent surgery using the strategies detailed in Table 2. The median follow-up was 68 months, the median DFS was 39 months (95% CI [12.71–65.29]), and the median OS was 66 months (95% CI [48.65–83.35]). The survival curves are shown in Figs. 3A and 3B. Of these, 26 underwent cervical lymph node dissection, and 19 did not. Median DFS was 61 months (95% CI [40.57–81.44]) for the dissection group and 17 months (95% CI [4.64–29.37]) for the non-dissection group. Median OS was 74 months (95% CI [57.54–90.46]) for the dissection group and 43 months (95% CI [35.90–50.09]) for the non-dissection group. The DFS and OS curves for both groups are shown in Figs. 4A and 4B. Both the median DFS and OS were significantly longer for the dissection group ($P = 0.049$ and $0.021$, respectively). The differences in median DFS and OS between the dissection group and non-dissection group were statistically significant ($P = 0.049$ and $P = 0.021$, respectively).

## Survival analysis by tumor site

In this study, based on the tumor origin, there were 45 cases in the parotid group, 14 in the submandibular group, and two in the sublingual group. Survival analysis showed no statistical differences in DFS and OS among SDC patients from different origins ($P = 0.071$, $0.44$, respectively). The DFS and OS curves are shown in Figs. 5A and 5B.

## Lymph node ratio as a prognostic factor

In our study, we utilized the lymph node ratio (LNR) as a metric to evaluate the impact of lymph node involvement on survival outcomes. Specifically, we analyzed 40 patients who underwent lymph node dissection. We performed survival analysis using Kaplan–Meier methods and Log-rank tests for univariate analysis. We systematically investigated various hypothetical critical values for LNR, ranging from 0.05 to 1.00 in increments of 0.05. The analysis revealed that when the LNR threshold was set at 0.15, the difference in DFS between patients with LNR $\leq 0.15$ and those with LNR $> 0.15$ was statistically significant

**Table 5  Comparison of median overall survival time among 45 parotid SDC patients with different clinical and pathological features.**

| Clinical and pathological features | | Median overall survival time (month) | (95% CI) | | P value | Log-rank |
|---|---|---|---|---|---|---|
| | | | Lower | Upper | | χ2 |
| Sex | Male | 63 | 45.62 | 80.38 | 0.79 | 0.07 |
| | Female | 51 | 23.04 | 78.96 | | |
| Age | <60 | 63 | 22.97 | 103.02 | 0.79 | 0.07 |
| | ≥60 | 61 | 38.61 | 83.39 | | |
| Surgical margin | Negative | 74 | 52.50 | 95.50 | 0.003 | 6.93 |
| | Positive | 25 | 18.54 | 31.46 | | |
| AJCC staging | I | | – | – | | 12.34 |
| | II | | – | – | | |
| | III | 79 | – | – | | |
| | IVA | 63 | 47.76 | 78.24 | | |
| | IVB | 37 | – | – | | |
| Pathological type | SDCexPA | 62 | 39.64 | 84.36 | 0.81 | 0.68 |
| | SDC *de novo* | 41 | 14.85 | 51.15 | | |
| PNI | Yes | 63 | 47.97 | 78.03 | 0.86 | 0.15 |
| | No | 54 | 26.42 | 81.58 | | |
| LVI | Yes | 37 | – | – | 0.72 | 0.20 |
| | No | 61 | 48.60 | 73.40 | | |
| ECE | Yes | 36 | 19.52 | 52.48 | 0.071 | 3.14 |
| | No | 51 | 27.45 | 74.55 | | |
| Treatment strategy | Surgery-chemoradiotherapy | 79 | 72.43 | 85.57 | | 9.83 |
| | Surgery-radiotherapy | 60 | 46.66 | 73.34 | | |
| Tumor diameter | <4 cm | 60 | 46.68 | 73.53 | 0.76 | 0.10 |
| | ≥4 cm | 63 | 31.58 | 94.42 | | |
| Lymph node dissection | Yes | 74 | 57.54 | 90.46 | 0.049 | 4.32 |
| | No | 43 | 35.90 | 50.09 | | |

Notes.

Abbreviation: SDCexPA, SDC ex pleomorphic adenoma; PNI, perineural invasion; LVI, lymphovascular invasion; ECE, extracapsular extension.

**Table 6  Multivariate survival analysis of 45 parotid SDC patients.**

| Factors | b-value | SE | Wald | P value | Exp (B) | 95% CI |
|---|---|---|---|---|---|---|
| Surgical margin | −0.99 | 0.41 | 5.830 | 0.015 | 0.37 | 0.17–0.82 |
| AJCC staging | −1.15 | 0.18 | 6.367 | 0.011 | 0.47 | 0.01–15.88 |
| Treatment strategy | 2.378 | 0.905 | 6.904 | 0.009 | 10.79 | 1.83–63.57 |
| Lymph node dissection | 0.878 | 0.447 | 3.858 | 0.04 | 2.40 | 1.00–5.78 |

($P = 0.016$). The DFS curves are shown in Fig. 6. However, for OS, none of the critical values tested yielded a statistically significant difference ($P > 0.05$).

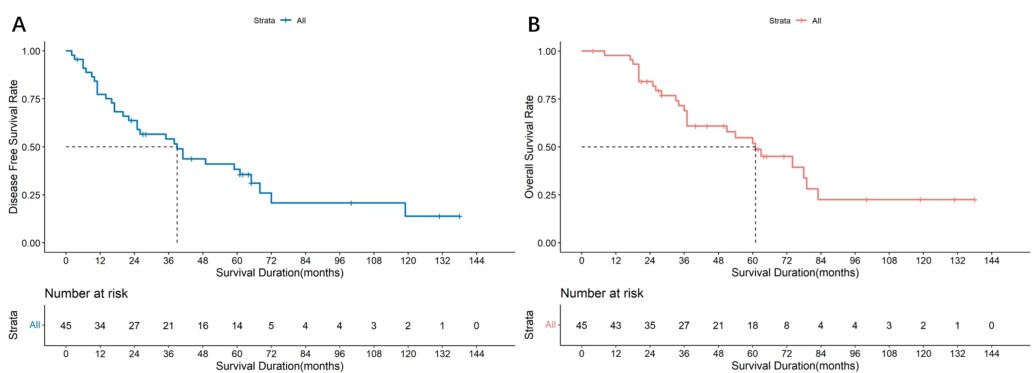

**Figure 3  Survival analysis of 45 parotid SDC patients.** (A) Disease free survival rate of 45 parotid SDC patients. (B) Overall survival rate of 45 parotid SDC patients.

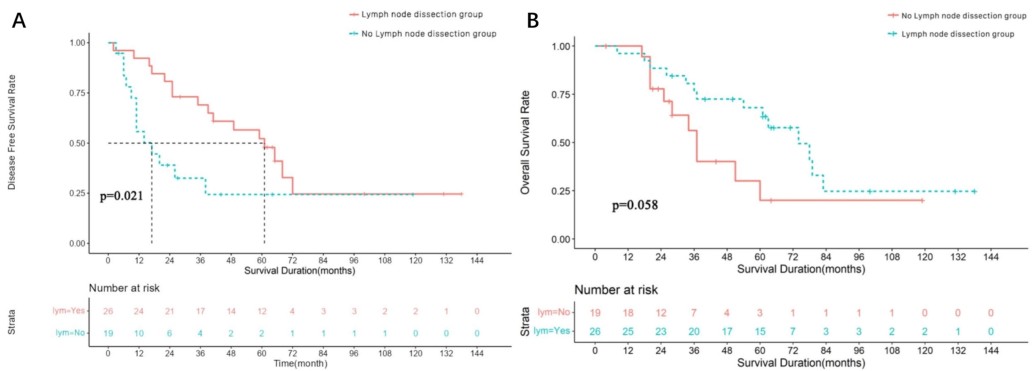

**Figure 4  Survival analysis of 45 parotid SDC patients undergoing lymph node dissection or not.** (A) Disease free survival rate of 45 parotid SDC patients undergoing lymph node dissection or not. (B) Overall survival rate of 45 parotid SDC patients undergoing lymph node dissection or not.

## DISCUSSION

SDC is a rare yet aggressive salivary gland malignancy characterized by a poor prognosis and a high propensity for regional lymph node metastasis. Our study aligns with prior research. SDC can develop *de novo* or arise from within a pleomorphic adenoma, with the latter constituting approximately 19.67% of cases in our cohort, a figure that falls within the range of 20%–59% reported in prior studies (*Chidananda-Murthy & Chandran, 2019*; *D'Heygere, Meulemans & Vander Poorten, 2018*). With an annual incidence of 0.04 per 100,000, SDC predominantly affects middle-aged and older men. In our study, the average age was 59.62 ± 11.22 years, and the male-to-female ratio was approximately 5:1, matching the literature's 4-5:1 ratio (*Aegisdottir et al., 2021*). As pathological diagnosis improves, SDC's proportion among SGCs is increasing; some previously classified SGCs might be SDC, suggesting its incidence is likely underestimated (*Goswami et al., 2020*; *Rooper et al., 2021*). The parotid gland had a significantly higher incidence (73.77%) than other body parts. Our findings stress considering SDC in middle-aged and older male

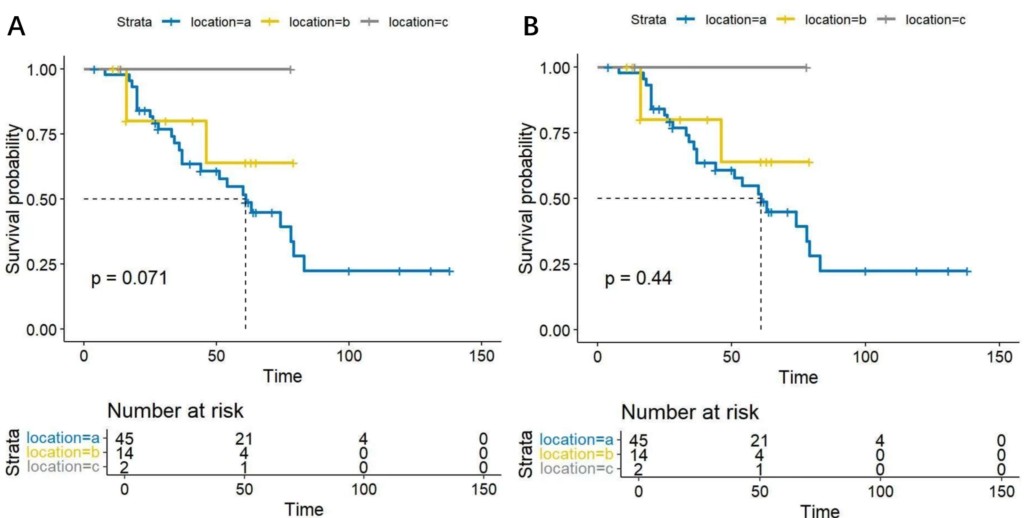

**Figure 5** (A) Disease free survival curves for SDC patients from different anatomical origins. The analysis showed no statistically significant difference s in DFS among SDC patients from different anatomical origins ($P = 0.071$). (B) Overall survival curves for SDC patients from different anatomical origins. The analysis revealed no statistically significant differences in OS among SDC patients from different anatomical origins ($P = 0.44$).

patients with parotid masses and highlight the need for ongoing vigilance in its diagnosis and management.

In our study, a significant proportion of patients presented with preoperative clinical symptoms indicative of malignancy, such as pain, facial nerve paralysis, a rapidly growing mass, and regional lymph node enlargement. These symptoms are consistent with the aggressive nature of SDC and its tendency to invade nerves and surrounding tissues, as described in the literature (*Enomoto et al., 2020*). The high rates of lymph node metastasis (57.38%) and distant metastasis (4.92%) in our cohort indicate that most SDC patients are diagnosed at advanced stages (III–IV), often with metastases in the lungs and bones. This aligns with previous reports (*Cruz et al., 2020*; *Zhang & Li, 2023*).

For most typical SDC cases, HE staining can confirm the diagnosis, while IHC offers deeper insights into prognosis and treatment. In our study, IHC analysis revealed high positivity rates for AR, GATA3, and GCDFP-15, which are regarded as diagnostic markers for SDC (*Chidananda-Murthy & Chandran, 2019*; *Jo et al., 2020*; *Nakaguro et al., 2020*). The positivity rates of AR and GCDFP-15 in our cohort were consistent with previous studies, which reported ranges of 67%–97% for AR and 66%–80% for GCDFP-15, while HER-2 expression varies significantly across different studies (*Chidananda-Murthy & Chandran, 2019*; *Egebjerg et al., 2021*; *Jo et al., 2020*; *Santana et al., 2019*). A meta-analysis of 3,372 cases of SGC reported an HER-2 positive rate of approximately 43% in SDC (*Egebjerg et al., 2021*). The median Ki-67 index in our study was 60% (range: 10%–85%), further indicating the high invasiveness of SDC (*Takase et al., 2017*). These molecular features highlight IHC's crucial role in diagnosing and prognosing SDC and the need for personalized treatment based on molecular markers.

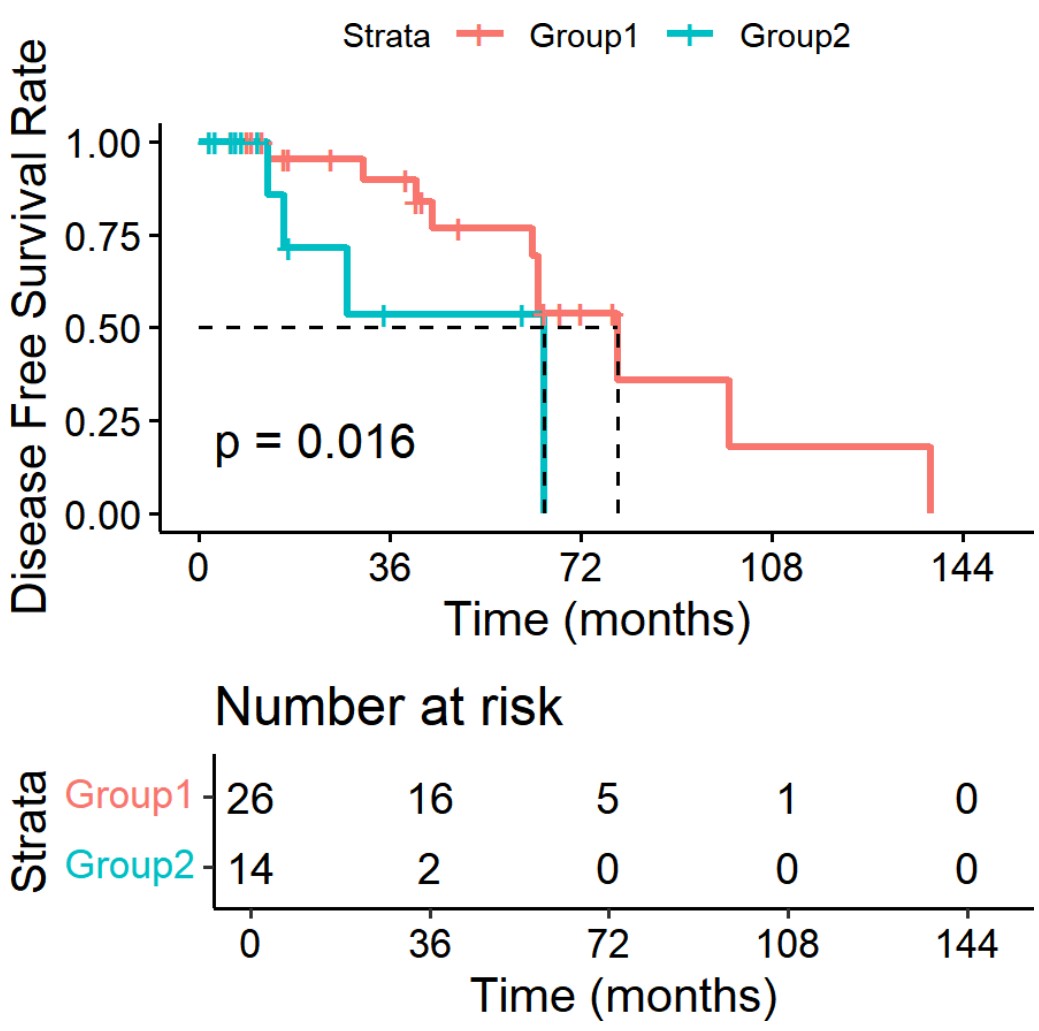

**Figure 6** **Disease free survival curves based on LNR in SDC patients.** Patients with LNR ≤ 0. 15 had significantly longer DFS than those with LNR > 0.15 ($P = 0.016$).

In addition to traditional detection methods, molecular detection methods are playing an increasingly important role in the diagnosis and treatment of SDC. By assessing the expression levels of PD-L1 and tumor mutation burden (TMB), we can evaluate a patient's potential response to immunotherapy (*Hirai et al., 2023*). For the detection of HER-2, further confirmation can be achieved through FISH, which is considered the "gold standard" for detecting gene amplification (*Von Mehren et al., 2022*). Next-generation sequencing technology can detect gene mutations, amplifications, and fusions in tumors, providing a more comprehensive basis for targeted therapy. The most frequent genetic alterations are mutations in TP53, HRAS, and PIK3CA; amplification of ERBB2; PTEN deletion; and BRAF pathogenic variants (*Dalin et al., 2016*; *De Lima-Souza et al., 2024*; *Mueller et al., 2020*; *Santana et al., 2019*). Our study highlights the importance of integrating molecular detection methods into the clinical management of SDC to better guide personalized treatment strategies.

According to the updated guidelines from the National Comprehensive Cancer Network (NCCN) and the European Society for Medical Oncology (ESMO), SDC treatment includes radical resection, lymph node dissection, and adjuvant therapy (*Caudell et al., 2022*; *Van Herpen et al., 2022*). Cervical lymph node metastasis is a key factor that influences the prognosis of SDC, therefore, some experts recommend cervical lymph node dissection, even for patients with a preoperative cN0 evaluation (*Nakaguro et al., 2020*; *Rahman & Griffith, 2021*). A study involving 22,653 patients with SGC showed that the lymph node metastasis rate for parotid SDC was approximately 54%, which was significantly higher than the rate for other types of SGC (approximately 24%) (*Xiao et al., 2016*). Another study by *Kusafuka et al. (2022)*, which included 304 SDC cases, found that 44% of cases involved nerve invasion, and 59% had positive lymph node metastasis. For patients with invasion of surrounding tissues, such as the facial nerve, lingual nerve, or mandible, extensive resection of the involved nerve, muscle, and bone tissue is recommended to achieve radical operation (RO) resection whenever possible (*Chidananda-Murthy & Chandran, 2019*).

In this study, 59 patients underwent radical resection, of whom 40 (67.80%) underwent lymph node dissection. Postoperative pathology revealed lymph node metastasis in 35 (55.75%) patients, PNI in 32 (54.24%) patients, and positive surgical margins in 17 (28.81%) patients. Among the nine cN0 patients who underwent cervical lymph node dissection, four (44.44%) showed metastasis on pathological examination. Of the 45 patients with parotid SDC, 26 (57.78%) underwent neck lymph node dissection, whereas 19 (42.22%) did not. The dissection group had a higher median DFS (61 *vs.* 17 months, $P = 0.049$) and median OS (74 *vs.* 43 months, $P = 0.021$). Among the 45 patients with parotid SDC, 17 (37.78%) underwent parotid tumor resection and superficial lobectomy, whereas 28 (62.22%) underwent total lobectomy. The latter group had higher median DFS (44 *vs.* 29 months, $P = 0.072$) and median OS (60.0 *vs.* 47.0 months, $P = 0.104$), though these differences were not statistically significant, possibly due to the small sample size.

Numerous studies support the use of postoperative radiotherapy for SDC, particularly in patients with high-risk factors, such as a tumor diameter greater than three cm, nerve involvement, positive surgical margins, and positive lymph node metastasis (*Rahman & Griffith, 2021*; *Scherl et al., 2019*; *Van Boxtel et al., 2019*). The NCCN guidelines recommend adjuvant radiotherapy for SDC, regardless of the tumor stage and resection margins (*Caudell et al., 2022*). Radical surgery plus adjuvant radiotherapy is beneficial for local control, but does not seem to control distant metastasis (*Kusafuka et al., 2022*). Consequently, some studies have explored the use of adjuvant chemotherapy, with platinum- and paclitaxel-based regimens becoming the most common choice for patients with recurrent or metastatic SDC, with an overall response rate of approximately 40%–50% (*Han et al., 2024*; *Imamura et al., 2022*). However, SDC generally responds poorly to chemotherapy, and whether combined chemotherapy can improve OS or reduce the rate of distant metastasis remains uncertain and requires further investigation (*Ho, 2021*; *Mimica et al., 2020*).

In this study, 52 (88.14%) patients received postoperative adjuvant therapy. Among them, 22 patients (42.31%) underwent chemoradiotherapy, while 30 patients (57.69%) received radiotherapy alone. Compared with patients treated with surgery alone, those who

received chemoradiotherapy or radiotherapy had improved survival, with the difference being statistically significant.

Personalized treatment based on molecular characteristics, such as AR and HER-2 have become possible for recurrent or metastatic SDC (*Mayer et al., 2024*; *Mueller et al., 2022*). The American Society of Clinical Oncology and ESMO guidelines recommend ADT for recurrent or metastatic SDC when AR expression exceeds 70% (*Geiger et al., 2021*; *Van Herpen et al., 2022*). Targeted therapies for AR and HER-2 have demonstrated good clinical efficacy in SDC (*Imamura et al., 2022*). Several retrospective studies have suggested that DFS significantly improves with androgen deprivation therapy (ADT) combined with radiotherapy in patients with AR expression greater than 70%, compared to radiotherapy alone (*Mueller et al., 2022*; *Uijen et al., 2020*; *Van Boxtel et al., 2019*). Compared to conventional chemotherapy, HER-2 targeted therapy offers better survival benefits, particularly for patients with locally advanced SDC or SDC with distant metastasis (*Kawakita et al., 2022*; *Sousa et al., 2022*). HER-2 targeted therapy combined with docetaxel has shown promising clinical efficacy in treating SDC, with an objective response rate of approximately 60%–72% (*Kawakita et al., 2022*; *Takahashi et al., 2019*). *Hanna et al. (2020)* also indicated that trastuzumab combined with concurrent chemoradiotherapy was beneficial for patients at high-risk, especially those with HER-2 3+ SDC.

In this study, two patients were found to have distant metastases at the time of diagnosis. One patient was treated with a combination of chemoradiotherapy and ADT, while the other received a combination of chemoradiotherapy, ADT, and trastuzumab. To date, these patients have survived for 11 and 14 months, respectively, with the tumor being controlled. An additional six patients received HER-2 targeted therapy after tumor progression following conventional chemoradiotherapy. During the follow-up period, four patients had stable disease, while two patients experienced disease progression.

Recently, monoclonal antibodies targeting Epidermal growth factor receptor (EGFR) have been explored for the treatment of advanced SDC. *Sato et al. (2024)* reported that a combination of an anti-EGFR monoclonal antibody (cetuximab) and paclitaxel showed good clinical efficacy in treating recurrent SDC. *Kawahara et al. (2017)* reported that a combination of cisplatin or 5-FU and cetuximab was effective in treating pulmonary metastases of SDC. Moreover, immunotherapy has shown promising results in various malignant tumors, and some researchers have attempted to use it for advanced SDC. *Ma et al. (2022)* performed comprehensive treatment with immune+targeted+chemotherapy (pembrolizumab+trastuzumab+cisplatin+capecitabine) in a case of advanced SDC of the submandibular gland with lung metastasis, achieving complete remission after six cycles. *Harwood et al. (2020)* reported that a patient with advanced parotid SDC achieved near-complete remission after receiving maintenance therapy with pembrolizumab for 2 years following multiple rounds of chemotherapy. These findings suggest that combination regimens could be viable options for some advanced SDC cases in clinical practice.

In this study, one patient with early stage parotid SDC received postoperative radiotherapy and anti-EGFR therapy (nimotuzumab), achieving a DFS of 62 months. Another patient with advanced parotid SDC who had high-risk factors such as positive surgical margins and lymph node metastasis, received combined radiotherapy and

chemotherapy after surgery. After disease progression and lung metastasis, the patient was treated with cetuximab and has survived with the disease for 29 months. A third patient with advanced SDC of the submandibular gland, and high-risk factors, including positive surgical margins and lymph node metastasis, could not tolerate the side effects of radiotherapy. After completing 11 fractions of radiotherapy, the patient received anti-EGFR therapy (cetuximab), and the disease remained stable. However, most of the treatments reported in the literature are based on case reports or small series of studies and lack confirmation through large-scale clinical trials. However, further prospective studies are required to validate the clinical efficacy of these treatments.

The prognosis for SDC is generally poor, with 5 year DFS and OS rates of 34.1%–41.0% and 42.0%–55.0%, respectively (*Hirai et al., 2023*; *Laughlin et al., 2023*). Our study found that surgical margins, AJCC stage, and treatment method were identified as adverse prognostic factors affecting the OS of patients with SDC. The impact of HER-2 and AR expression on prognosis varies across different studies. Some studies have suggested that the positive expression of HER-2 and AR is associated with reduced OS and DFS in patients with SDC (*Mueller et al., 2022*; *Santana et al., 2019*). However, other scholars have argued that their expression is not related to tumor invasiveness or prognosis (*Di Villeneuve et al., 2021*; *Han et al., 2024*; *Williams et al., 2023*). Recently, the significance of HER-2, AR in SDC has generated considerable interest, particularly for the adjuvant treatment of SDC. Studies on SDC have demonstrated that the AR signaling pathway is involved in tumor progression and invasiveness. In addition to directly killing AR-positive cancer cells *via* apoptosis and prolonging the doubling time by actively dividing cells into a quiescent state, ADT influences tumor angiogenesis (*Dalin et al., 2017*). HER-2 protein is a transmembrane glycoprotein with tyrosine kinase activity. Its overexpression can potentially activate the signaling pathways of the epidermal EGFR and promote tumorigenesis. Some AR-positive SGCs, particularly SDC, simultaneously express HER-2. In breast cancer, HER-2 blockade may improve the response to endocrine therapy, because the interaction between HER-2 and hormone receptor pathways can lead to endocrine therapy resistance (*Johnston et al., 2021*). Therefore, a similar interaction between HER-2 and AR may also exist in SDC.

Tumor heterogeneity and variations in clinical practice may influence outcomes. The heterogeneity of SDC, including variable histological features and expression of molecular markers such as AR, HER-2, and EGFR, may impact treatment strategies and prognosis. This heterogeneity necessitates a personalized approach to therapy. Additionally, variations in clinical practice, such as differences in surgical techniques, adjuvant therapy indications, and follow-up schedules, can influence treatment outcomes. Standardizing clinical protocols and using molecular profiling to guide treatment decisions may help optimize therapeutic efficacy and improve patient prognosis.

This study enrolled 61 patients with SDC, making it one of the largest reported studies to include Chinese patients. This is of significant importance for enriching the global research data on SDC. The findings of this study provide valuable insights into the diagnosis and treatment of SDC. However, this study has some limitations. First, the low incidence of SDC and the characteristics of single-center, retrospective studies make it difficult to eliminate selection bias. Second, the lack of a standardized adjuvant regimen for SDC may

have impacted the assessment of its effectiveness in improving survival rates. Third, due to a variety of factors, some patients did not undergo neck lymph node dissection, which may have had an impact on the outcomes.

## CONCLUSION

SDC is a rare and highly malignant tumor in clinical practice. While advances in treatment modalities such as surgery, radiotherapy, chemotherapy, and targeted therapies have improved our ability to manage SDC, survival rates remain low. The aggressive nature of SDC, characterized by rapid progression, early metastasis, and resistance to conventional therapies, poses significant challenges to improving survival outcomes. Additionally, the heterogeneity of SDC tumors, with variations in molecular markers such as AR, HER-2, and EGFR, can lead to differing responses to therapy, making it difficult to establish uniform treatment protocols. The integration of molecular targeted therapies and immunotherapy offers promising avenues for personalized treatment, but their clinical efficacy remains to be validated in larger prospective studies. With the advancement of molecular detection technologies and the development of more targeted drugs, precision medicine will play an increasingly important role in the treatment of SDC. However, significant challenges remain in improving the prognosis of patients with this aggressive malignancy. Ongoing research is essential to further explore the molecular mechanisms underlying SDC pathogenesis and to identify novel therapeutic targets and biomarkers. Additionally, there is a critical need for large-scale, multi-center studies to validate the clinical efficacy of emerging treatments and to refine personalized treatment approaches. The continuous evolution of research and innovation will be vital in overcoming the current limitations in SDC therapy and in ultimately improving patient outcomes.

## ACKNOWLEDGEMENTS

We would like to thank Editage for English language editing.

### Funding

The authors received no funding for this work.

### Competing Interests

The authors declare there are no competing interests.

### Author Contributions

- Shubin Dong conceived and designed the experiments, prepared figures and/or tables, authored or reviewed drafts of the article, and approved the final draft.
- Mengru Li performed the experiments, analyzed the data, prepared figures and/or tables, and approved the final draft.
- Zhiwei Zhang conceived and designed the experiments, performed the experiments, prepared figures and/or tables, and approved the final draft.

- Bowei Feng conceived and designed the experiments, performed the experiments, authored or reviewed drafts of the article, and approved the final draft.
- Wei Ding performed the experiments, analyzed the data, prepared figures and/or tables, and approved the final draft.
- Jiang Chang conceived and designed the experiments, analyzed the data, authored or reviewed drafts of the article, and approved the final draft.
- Feng Liu conceived and designed the experiments, analyzed the data, authored or reviewed drafts of the article, and approved the final draft.

## Human Ethics

The following information was supplied relating to ethical approvals (i.e., approving body and any reference numbers):

This study has received approval from the Medical Ethics Committee of Shanxi Cancer Hospital (Ethics number: KY2024081), and informed consent has been duly obtained from both the patients and their families.

## Data Availability

The raw measurements are available in the Supplemental Files.

## Supplemental Information

Supplemental information for this article can be found online at http://dx.doi.org/10.7717/peerj.19626#supplemental-information.

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
