# Peer review of "Clinical diagnosis, treatment, and survival analysis of 61 cases of salivary duct carcinoma: a retrospective study"

_PeerJ, doi:10.7717/peerj.19626_

## Round 0.1 · original submission · Major Revisions

Your article requires major revisions. In particular, you must address the concerns of Reviewer 2 in detail

Reviewer 1 ·

Basic reporting

This single-institution series of salivary duct carcinoma is overall well written. Figures and data tables are clear. However, there is a lot of redundant information about SDC in the Introduction and Discussion. I would suggest making the Intro much more concise, and leaving the review of the literature for the Discussion. The Discussion is also too lengthy; consider dramatically consolidating the paragraphs on systemic therapy for SDC. A comprehensive review of that subtopic is not needed here.

Overall, I also think it is necessary to emphasize what this particular study adds to the literature. There are numerous single-institution series of SDC, many of them larger than this one. I do think this may be one of the largest series including Chinese patients. If so, please emphasize that.

Additional minor points:
1) Lines 257-259: There is a response rate quoted for adjuvant chemo, but responses to chemo are not applicable in the adjuvant setting. Please clarify.

Experimental design

The statistical methods are sound. One suggestion- why not include HER2 expression as a variable in the statistical analyses, since you have the data?

Other minor points:
1) It is stated that 57% had lymph node metastases- was this according to imaging or pathology? Assuming imaging, since neck dissection was performed on just over half of the parotid cases?
2) How was treatment decided? Why did some patients get chemoRT whereas others just RT?

Validity of the findings

The authors acknowledge the lack of standardized treatment at their institution as a limitation. However, they under-emphasize the extent to which their treatment paradigms may have affected prognosis. For example, why was neck dissection performed in just half of the cases, when it is well established (in articles cited here) that the rate of occult metastasis, even in clinically N0 cases, is as high as 50%? They later state that neck dissection is recommended for all cases, but failed to emphasize that they themselves did not do this, and this may have led to substandard outcomes.

Reviewer 2 ·

Basic reporting

Grammar & Clarity: Review for grammatical and typographical errors. Ensure acronyms are spelled out the first time they are used.
Title & Abstract: The title is clear but could be more specific by mentioning survival analysis and adjuvant therapy. Clarify certain sentences in the abstract for conciseness and remove irrelevant details, such as the sentence about accessing patient data. The conclusion could be more assertive.
Data Presentation: Ensure tables and figures are cross-referenced in the text, and clarify the categorization of data (e.g., sex vs. gender, inclusion of pathologic data).

Experimental design

Study Design: Clarify the study design and inclusion/exclusion criteria. Justify the time frame for case selection and the rationale for adjuvant treatments.
Methodology: Explain diagnostic methods and criteria for lesions. Provide more details on preoperative exams and surgical decisions. The role of the "dedicated research nurse" in follow-up needs further clarification.
Statistical Methods: Specify statistical tests used, especially for Cox models. Provide more statistical details in the results, such as confidence intervals and p-values.

Validity of the findings

Interpretation of Results: The pathologic data presented are sparse. Histologic classifications, tumor grading, and other relevant factors such as capsular extension should be included.
Clinical Significance: The relevance of AR and HER-2 findings should be more thoroughly explained in terms of their impact on treatment options. Discuss the use of molecular markers as part of the study’s novelty.
Replicability: Methods used for lesion diagnosis, tumor classification, and statistical analysis should be detailed enough for replication.

Additional comments

This study addresses an important topic and has great potential to contribute to the field of head and neck pathology. However, after careful review, I believe that certain aspects of the manuscript would benefit from further refinement to enhance its clarity and overall impact. Additional information regarding inclusion and exclusion criteria and pathologic characteristics would provide more context for the study. Given the relevance of pathological characteristics to prognostic and therapeutic aspects, addressing these points would strengthen the manuscript. Addressing these issues will significantly improve the quality of the manuscript and its potential contribution to the field of head and neck pathology.

General Comments
• Check for grammatical and typographical errors.
• Review all acronyms and make sure they are spelled out the first time they are used, e.g., AR, CT, MRI, etc.
• Adding a pathologist to the team can increase the robustness of the data description and broaden the perspectives for data analysis.

Title and Abstract
• The title and aim are clear but could be made more specific by mentioning the focus on survival analysis and adjuvant therapy.
• The sentence "On May 21, 2024, we accessed the patient data..." is irrelevant and could be omitted. Replace with a concise statement: "The data were retrospectively collected from patients admitted to Shanxi Cancer Hospital between January 2010 and February 2024."
• Including information about inclusion and exclusion criteria would improve clarity.
• Use more concise phrasing to avoid information overload. For example, combine treatment group and survival data in a simplified statement: "Median overall survival was significantly higher in the adjuvant treatment groups (radiotherapy or chemoradiotherapy) compared with the surgery only or nonsurgery groups (P < 0.05)."
• Briefly explain the relevance of the immunohistochemical findings (AR and HER-2 positivity) for clinical management.
• The conclusion could be more assertive in emphasizing the importance of adjuvant therapy for survival: "Radical surgery remains the primary treatment for salivary duct carcinoma, while adjuvant therapy significantly reduces recurrence rates and improves survival. However, the overall prognosis remains challenging, highlighting the need for novel therapeutic strategies".

Introduction
• Reorganize the introduction to follow a clear logical progression: definition, epidemiologic, clinical, pathologic, molecular, and prognostic features, and treatment gaps.
• The term "highly malignant tumor" should be clarified. If it refers to "aggressive" or "high-grade", rephrase accordingly.
• The histologic and etiologic basis of SDC is confusing. Simplify by stating: "SDC arises from the epithelial cells of the salivary duct excretory system and may develop de novo or from a pre-existing pleomorphic adenoma (SDC ex-PA)."
• Explain the impact of molecular approaches to treatment and how this distinguishes the study.
• Confirm the relevance of the references cited and consider including more recent literature (e.g. WHO 2022).
• Specify the aim of the study and relate it to the gaps discussed.

Materials and Methods
• The study mentions that the data was accessed on "May 21, 2024," which seems inconsistent with the follow-up ending in "June 2024. Clarify the timeline.
• What were the inclusion and exclusion criteria for the study?
• It is unclear if the 61 patients represent a consecutive sample or if additional selection criteria were applied. This could introduce bias.
• Why were 2010 and 2024 chosen as the time frame for case selection? Justify this choice.
• The description of preoperative examinations is detailed, but references or justifications for the use of CT, MRI, and ultrasound together are lacking.
• Provide details on how cases were assigned. Was a prior biopsy performed?
• Explain how lesions were diagnosed and the criteria used. There are no details on the pathologic characteristics or diagnostic methods of the lesions.
• The surgeon's decision is vaguely described as "based on a comprehensive evaluation". Add objective criteria (e.g. surgical margins) for intraoperative decisions to improve reproducibility.
• The follow-up process is clearly described, but the role of the "dedicated research nurse" needs to be better contextualized. Was specific training provided? Was patient follow-up consistent?
• The mention of "Figure 1" suggests data visualization, but its content is not described in the text. In addition, Figure 1 is too cluttered.
• The rationale for adjuvant treatment choices is missing. Why did some patients receive radiation therapy and others did not?
• Was HER-2/AR status evaluated as a guide for adjuvant therapy? If so, at the protein or genetic level? What methods and criteria were used? Were NOS adenocarcinoma cases tested for AR?
• Statistical methods: Were covariates for the Cox model prespecified or identified by univariate analysis? The statement "all data follow a normal distribution" is vague. Specify what tests were performed.

Results
• Include cross-references to tables and figures in the text (e.g., "As summarized in Table 1," "The survival curves are shown in Figure 2B") to help readers navigate the data.
• Remove the term "stage" in the T, N, and M columns of Table 1.
• Pathologic data are sparse. How were the cases classified histologically? Was tumor grading, growth pattern, necrosis, or cell proliferation index assessed? Specify capsular extension for SDC ex-PA cases.
• Replace "sex" with "gender".
• For three patients with distant metastases (M1), indicate the sites of metastases.
• Briefly explain why certain patients received different treatments, such as surgery combined with radiation or chemotherapy.
• Clarify whether factors such as tumor stage or surgical margins influenced adjuvant therapy strategies.
• Condense redundant discussions of OS and DFS in the sections describing cervical dissection.
• Provide more statistical details (e.g., confidence intervals and p-values) to strengthen the results.
• Justify the focus on parotid cases. Were there unique characteristics that warranted this separation? Instead of splitting data by site, consider survival analysis by tumor site and compare de novo SDC vs. SDC ex-PA cases.
• For patients undergoing neck dissection, include an analysis of the number of positive vs. negative lymph nodes as an additional prognostic factor.

Discussion
• The paragraphs are long and hard to read. Break them up to improve clarity and flow.
• The authors discuss findings (e.g. immunohistochemical and molecular markers) that are not included in the Results section. Ensure that these findings are presented in the Results along with their methodology.
• Expand on how AR and HER-2 findings affect treatment options and outcomes.
• Reorganize the treatment discussion to present standard therapies first, followed by emerging approaches such as targeted and immunotherapies.
• Discuss how tumor heterogeneity and variations in clinical practice may have influenced the results.

Conclusion
• Mention the need for collaborative research efforts or the role of precision medicine.
• Emphasize the aggressive nature of the disease and the difficulty in improving survival rates despite advances in treatment.
• End with a forward-looking statement emphasizing the importance of further research and development.

Reviewer 3 ·

Basic reporting

This is a single-center, retrospective cohort study using survival analysis. A retrospective analysis of the treatment and prognosis of salivary ductal carcinoma (SDC) patients were undertaken, altogether 61 cases SDC were enrolled which had completed treatment and follow-up data in this institution in a long period time (about 14 years). Most of the patients were at advanced cancer stage at the time of consultant, radical surgery was still the first choice of treatment, and postoperative chemo- or chemoradiotherapy was helpful to prolong the patients’ survival time, but the overall prognosis was still far from optimistic. A separate subgroup survival analysis was also performed for 45 cases of parotid ductal carcinoma.
The introduction of the research background is sufficient, and the reference is selected and cited properly. The English language should be improved to ensure that an international audience can clearly understand the text. I suggest the authors have someone who is proficient in English and familiar with the subject matter give someone writing tips of the manuscript.

Experimental design

Indeed, the design of the study has some illogical and defects need to be fixed.
First of all, it is indicated in this paper that a long-term follow-up was conducted on this group of cases (Line 104-112), and all patients had signed their informed consent (Line 77-79). Please clarify whether this study is a prospective study or a retrospective study, a cross-sectional study or a cohort study, and express this clearly in the manuscript.
Univariate and multivariate survival analysis were used in statistical methods to evaluate the impact of clinicopathological variables on overall survival (OS) and disease-free survival (DFS). However, the interpretation and statement of K-M curve and Cox proportional risk model results were not correct(line 134-139, line149-151, line157-159,line162-171 ). The table was not a formal table for univariate and multivariate analysis (table 3,4,5,6), so they are all needed to be modified. The log-rank test result needs to be marked on Figure 3,6,7. Please note whether the specific value expressed in the text is a 95% confidence interval or a range, and then correct it

Validity of the findings

no comment

Additional comments

no comment

Annotated reviews are not available for download in order to protect the identity of reviewers who chose to remain anonymous.

·

Basic reporting

Line 125 - you mention a number of patients had symptoms of nerve invasion - can you elaborate what type of symptoms you are referring to? Paresthesia/anesthesia/palsy/ etc. I would also elaborate on which nerve (i.e. facial nerve, IAN, etc) was involved

Line 126 You mention that nerve invasion was observed in 32 patients (54.24%) on postoperative pathology examination - I would keep this consistent and say perineural invasion (PNI) to prevent any confusion.

Experimental design

Were all surgeries performed by the same surgeon? If not, might be a good idea to indicate how many surgeons involved, level of experience, specialty, etc.

Validity of the findings

Line 127 and 128 you mention that 3 of the patients had distant mets, but in lines 142 and 143 and in other areas throughout the article you mention that only 2 of the patients received palliative chemoradiotherapy due to distant metastasis. Is this a typo?

---

## Round 0.2 · Major Revisions

Dear authors,

Unfortunately, the concerns raised previously have not been properly addressed and/or clarified, incl. in the manuscript itself.

Pathological data is essential for understanding prognosis and treatment outcomes (and is lacking which compromises certain conclusions and affirmations). Make sure that key methodological details , including inclusion / exclusion criteria and study timeline are crystal clear!

Also, there seems to be major issues in data interpretation and statistical reporting,... and/or coming from persistent writing and language deficiencies that affect clarity and readability.

Moreover, despite the importance of the topic and rarity of the condition you should uphold scientific rigor! Please, refer to the actual and previous reports of the reviewers, particularly reviewer 2.

Reviewer 1 ·

Basic reporting

No concerns

Experimental design

No concerns

Validity of the findings

No concerns

Additional comments

The authors have addressed my concerns.

Reviewer 2 ·

Basic reporting

Professional, clear, and unambiguous English is used throughout the text. There are still errors in the text.

Professional article structure - Results

Experimental design

Methods are described with insufficient detail

Validity of the findings

The study is not groundbreaking.

Additional comments

While the authors have made some modifications, several important issues remain unaddressed. Some of these concerns were already highlighted in the previous review, yet they have not been fully resolved in the revised version.

Introduction
1. Clearer explanation of the impact of molecular approaches on treatment - The text mentions the use of personalized therapies for AR and HER-2, but does not explain how these approaches differentiate the study from others in the field. It would be interesting to highlight how this study specifically contributes to the field.
2. Aim of the study in relation to the discussed gaps - The aim is mentioned at the end, but is still somewhat general. It would be better to explicitly relate it to the gaps mentioned, such as the need for new treatments and improved patient survival.

Materials and Methods
1. Inconsistency in timeline - The text mentions that data were collected until February 2024, but the follow-up ended in August 2024. This is still confusing because it doesn't explain how the data were accessed before the end of the follow-up.
2. Justification for study period (2010-2024) - Addressed in the response letter, but not in the manuscript.
3. Use of imaging - No justification for the combined use of ultrasound, tomography, and magnetic resonance imaging. Guidelines or literature supporting this approach must be cited.
4. Diagnostic criteria and pathological methods - The text mentions the use of the WHO 2022 classification, but does not describe in detail which histological and immunohistochemical features were evaluated.
5. Inclusion criteria: Present in body text rather than bullet points. Indicate that both new and ex-pleomorphic adenoma cases were included. Exclusion criterion 4 seems unnecessary, as it would show the true landscape of SDC treatment.
6. Lack of correlation between survival analysis and pathologic features
7. "In this study, three senior professors of surgery participated in the surgical treatment with assistants, all from the hospital's Department of Head and Neck Surgery. These three professors had over 15 years of clinical work experience and extensive surgical expertise in salivary gland tumors." This entire statement is irrelevant and could be simplified to "All cases were operated on by surgeons experienced in salivary gland tumor surgery.
8. "In addition, some patients underwent further imaging studies, such as cervical computed tomography or magnetic resonance imaging, according to specific conditions." What specific conditions?
9. Temporality: Some sentences in the Methods remain in the present tense when they should be in the past tense.
10. Rationale for adjuvant therapy - It is still unclear why some patients received radiation therapy while others did not. Were biomarkers such as HER-2 and AR considered to guide this decision?
11. HER-2 and AR status - Addressed in the reply letter but not in the manuscript.
12. Cox model and statistical assumptions - The text mentions that the data follow a normal distribution, but does not specify what test was used to verify this. It is also unclear whether the covariates in the Cox model were pre-selected or selected after univariate analysis.

Results
1. The results do not follow a logical order. The topic "Basic Information and Treatment Strategies for Patients with SDC" includes information that fits into other topics (e.g., pathologic - PNI, LVI, origin - de novo vs. ex-pleomorphic adenoma, grade, immunohistochemistry...). I propose to divide it into: 1. Clinical and epidemiologic characteristics (sex, age, location, progression, symptoms, clinical TNM, etc.); 2. Pathologic characteristics (everything that can be assessed with histologic analysis material, such as classification - de novo vs ex-pleomorphic adenoma, PNI, LVI, origin - de novo vs ex pleomorphic adenoma, grade, immunohistochemistry, etc.); 3. Treatment, tumor behavior, and follow-up (therapeutic modality, recurrence, metastasis, metastasis location - addressed in the response letter but not in the manuscript -, follow-up, patient status...).
2. Detailed pathological data: The text now includes information on histologic grade, extracapsular extension, perineural and lymphovascular invasion. It also mentions HER-2 gene amplification and other molecular features. However, there is no clear mention of growth pattern, necrosis, or cell proliferation index, described in the response letter but not in the text.
3. The correct term is sex (biological), not gender (based on social constructs). Apologies for the error in the previous review.
4. Explain differences in treatment (surgery + radiotherapy/chemotherapy): The text now mentions different treatment groups and their median survival, but does not explain the criteria that led certain patients to receive chemotherapy/radiotherapy in addition to surgery - leave the explanation in the methodology.
5. Justify the focus on parotid cases and consider survival analysis by tumor site: The text maintains the separation of parotid cases without clearly justifying this choice, addressed in the letter, but not in the text.
6. Include analysis of positive versus negative lymph nodes as a prognostic factor: The text now mentions that 26 patients underwent cervical dissection and that survival was higher in this group. However, there is no explicit analysis of the number of positive vs. negative lymph nodes - explained in the response letter, but not in the text.

Discussion
1. The discussion could be better written and correlated, rather than simply taking data from the literature and then from the paper.
2. Long Paragraphs - The revised text still contains long paragraphs, which may hinder readability and clarity. Some sections could be split to improve flow.
3. Reorganization of Treatment Section - The structure of the treatment section still mixes conventional and emerging therapies. It would be clearer to first present standard approaches (surgery, radiotherapy) and then discuss emerging targeted therapies and immunotherapy.
4. Discussion of tumor heterogeneity and variations in clinical practice - The text mentions differences in prognostic findings related to AR and HER-2, but there is no explicit discussion of how tumor heterogeneity and variations in clinical practice may influence the results. This consideration could be further elaborated.
5. "The commonly mutated genes in SDC include BRAF, PIK3CA, and TP53 (Mueller et al., 2020)". The most frequent genetic alterations are mutations in TP53, HRAS, and PIK3CA; amplification of ERBB2; PTEN deletion; and BRAF pathogenic variants (10.1158/1078-0432.CCR-16-0637, 10.1016/j.humpath.2019.08.009, 10.1038/s41379-020-0576-2, 10.1097/PAS.0000000000002307).

Conclusion
1. The text mentions that survival rates are low, but could better emphasize the difficulty of improving them despite advances in treatment.
2. The last sentence talks about the need to develop new therapeutic targets and biomarkers, which is good, but could be more impactful to reinforce the ongoing importance of research and innovation.

Reviewer 3 ·

Basic reporting

This is a single-center, retrospective cohort study using survival analysis. A retrospective analysis of the treatment and prognosis of salivary ductal carcinoma (SDC) patients was undertaken; altogether, 61 cases of SDC were enrolled, which had completed treatment and follow-up data in this institution over a long period of time (about 14 years). The revised manuscript explained and modified the suggestions made earlier. The writing language of the manuscript is reasonable and clear.

Experimental design

-

Validity of the findings

-

Additional comments

There are still some issues that need to be mentioned.
1. Generally, when comparing variables between two groups of A and B, if the P-value inferred from the calculated statistic is less than 0.05, it should be expressed as: the difference between A and B is with statistical significant, rather than as “there is a significant difference between A and B”; Similarly, if the P-value is greater than 0.05, it is stated that the difference between A and B is not with statistical significant, and it is considered that there is no difference between A and B, rather than that “there is no significant difference between A and B”
In the RESULT Section of this manuscript (like Line187-190, Line204-206, Line208-211, Line223-225...), some expressions were still not correct. I suggest the authors consult some relevant statistical experts to make some corrections.
2. For Tables 3 and 4, it is necessary to note the statistical methods used for inter-group comparison and the calculated statistics, and both the statistics and P-values should be listed in the table
3、Line 195 “Among the 61 patients, the median follow-up time was 64 months (95%CI: 60.28–67.72) …”, Line 217 ”The median follow-up was 68 months (95%CI: 57.29–78.71)…. Time of follow-up is an observational variable, and 95%CI do not need to be calculated.

·

Basic reporting

-

Experimental design

-

Validity of the findings

-

Additional comments

Thank you for addressing my concerns.

---

## Round 0.3 · Minor Revisions

Dear authors,

Thank you for your endeavours. A few minor yet important issues remain, particularly in terms of methodology description or reporting, that must be resolved before the manuscript can be considered for publication. Please refer to the reviewer's report.

Reviewer 2 ·

Basic reporting

Acceptable

Experimental design

Verbal inconsistency

Validity of the findings

Acceptable

Additional comments

General Comments:
The authors addressed most of the previous comments, resulting in significant improvements to the clarity and structure of the manuscript. However, a few minor but important issues remain that must be resolved before the manuscript can be considered for publication.

Materials and Methods:
The authors stated that data collection was conducted until February 2024. However, they contacted patients again in August to obtain follow-up information. Therefore, data collection effectively continued until August 2024, which should be clearly indicated in the manuscript.
Although most of the text has been revised to use the past tense, some parts are still written in the present. For example: "Ultrasound, CT, and/or MRI are combined..." and "the extent of resection is determined...". For consistency and clarity, it is important to standardize all verb tenses to the past in the Methods section.
In the response letter, the authors addressed the "specific conditions" for image usage. However, these conditions are not mentioned in the manuscript itself. To ensure the study's reproducibility, the authors should include these specific conditions in the text.

Results:
The subtitles "The clinical, pathological, and survival analysis of 61 cases of SDC" and "The clinical, pathological, and survival analysis of 45 cases of parotid SDC" should be restructured as subheadings under a broader section, such as "Statistical Results" or "Survival Analysis." To avoid repetition, I recommend revising these subtitles to something more concise, such as "All cases" and "Parotid cases."

Discussion:
The discussion section is excessively long, with extended paragraphs that tire the reader. The authors are encouraged to condense and reorganize this section to enhance readability and maintain a more fluid narrative.

---

## Round 0.4 · accepted · Accept

Dear authors,

Congratulations! We have reached an acceptance decision. I ask you to be thorough in your proofreading of this manuscript and wish you good luck for future investigations.

Reviewer 2 ·

Basic reporting

No comment

Experimental design

No comment

Validity of the findings

No comment

Additional comments

While improvements have been made in paragraph structuring and overall flow, the discussion section remains overly long. Further condensation—especially in treatment-related segments—by reducing repetition and focusing more on interpretation rather than reiteration of results and literature data, could significantly enhance readability.